# Integrated Logic Circuits Based on Wafer-Scale 2D-MoS_2_ FETs Using Buried-Gate Structures

**DOI:** 10.3390/nano13212870

**Published:** 2023-10-30

**Authors:** Ju-Ah Lee, Jongwon Yoon, Seungkwon Hwang, Hyunsang Hwang, Jung-Dae Kwon, Seung-Ki Lee, Yonghun Kim

**Affiliations:** 1Department of Energy and Electronic Materials, Surface Materials Division, Korea Institute of Materials Science (KIMS), Changwon 51508, Republic of Korea; jualee0208@kims.re.kr (J.-A.L.); jwyoon@kims.re.kr (J.Y.); hhs0505@kims.re.kr (S.H.); jdkwon@kims.re.kr (J.-D.K.); 2School of Materials Science and Engineering, Pusan National University, Busan 46241, Republic of Korea; 3Center for Single Atom-Based Semiconductor Device, Department of Materials Science and Engineering, Pohang University of Science and Technology (POSTECH), Pohang 37673, Republic of Korea; hwanghs@postech.ac.kr

**Keywords:** molybdenum disulfide, buried-gate structure, logic circuits, gate controllability, wafer-scale integration

## Abstract

Two-dimensional (2D) transition-metal dichalcogenides (TMDs) materials, such as molybdenum disulfide (MoS_2_), stand out due to their atomically thin layered structure and exceptional electrical properties. Consequently, they could potentially become one of the main materials for future integrated high-performance logic circuits. However, the local back-gate-based MoS_2_ transistors on a silicon substrate can lead to the degradation of electrical characteristics. This degradation is caused by the abnormal effect of gate sidewalls, leading to non-uniform field controllability. Therefore, the buried-gate-based MoS_2_ transistors where the gate electrodes are embedded into the silicon substrate are fabricated. The several device parameters such as field-effect mobility, on/off current ratio, and breakdown voltage of gate dielectric are dramatically enhanced by field-effect mobility (from 0.166 to 1.08 cm^2^/V·s), on/off current ratio (from 4.90 × 10^5^ to 1.52 × 10^7^), and breakdown voltage (from 15.73 to 27.48 V) compared with a local back-gate-based MoS_2_ transistor, respectively. Integrated logic circuits, including inverters, NAND, NOR, AND, and OR gates, were successfully fabricated by 2-inch wafer-scale through the integration of a buried-gate MoS_2_ transistor array.

## 1. Introduction

Graphene, which consists of an atomic thickness arrangement of carbon atoms in a hexagonal lattice, was discovered by Andre Geim and Konstantine Novoselov in 2004 [1,2,3]. Since then, in recent years, graphene has received significant attention as a revolutionary material owing to its excellent mechanical strength and remarkable electrical conductivity, which presents exciting potential for next-generation electronic devices. Furthermore, graphene’s high electron mobility promotes fast electron movement with minimal scattering, offering advantages for high-speed electronic devices [4,5,6]. While graphene lacks intrinsic bandgap, semiconductors usually have a bandgap that allows them to switch between conducting and insulating states. However, the band structure of graphene shows linearity that offers excellent conductivity, but the on/off ratio of graphene-based transistors can be relatively low. Thus, it is difficult to create the off-state and it can cause problems when trying to precisely control current flow, which is critical for digital logic operation. However, there are actively researched approaches for new materials, such as two-dimensional (2D) transition metal dichalcogenides (TMDs) that possess semiconductor properties, thus addressing certain limitations of graphene [7,8,9,10,11]. Also, high-performance organic transistors have been pursued [12,13,14].

Among them, two-dimensional (2D) transition metal dichalcogenides (TMDs) materials, such as molybdenum disulfide (MoS_2_), tungsten disulfide (WS_2_), tungsten diselenide (WSe_2_), and others, are attracting attention as promising materials for next-generation electronic devices [15,16]. Two-dimensional (2D) transition metal dichalcogenides (TMDs) materials are single-layer atomic structures [17,18] exhibiting excellent stability and a wide range of properties due to strong covalent and Van der Waals forces [15,19,20,21]. In addition, they stand out as high-level electrical properties such as tunable band gaps, low power consumption, high field-effect mobility, and switching characteristics owing to the high on/off ratio [22,23]. Therefore, it shows a wide range of applications in the various fields. For example, previous reports on photodetectors [24], optoelectronic memories [25], sensors [26], flexible electronic devices, and logic devices are currently being actively studied [7,27,28,29,30,31].

Specifically, 2D semiconductors such as molybdenum disulfide (MoS_2_) can be fabricated in high-quality electronic-grade quality on a wafer-scale [27,31,32]. The MoS_2_ could provide the ultra-thin body of sub-1 nm with an intrinsic flat surface which is immune to the short-channel effect. Moreover, high-electron mobility comparable to the silicon channel has been reported with the development of the chemical vapor deposition (CVD) or metal-organic chemical vapor deposition (MOCVD) growth technique. Therefore, it is a promising 2D material for the potential perspective of more Moore and more than Moore pushing technology [16,33,34,35,36,37,38]. When considering device architectures, several critical factors determine device performance, including contact resistance, dielectric scaling, and conformal gate controllability. In general, the MoS_2_ field-effect transistor (FET) based on a local back-gate structure results in non-uniform distribution of the gate electric field. This is caused by an abnormal gate shape that occurs during the conventional lift-off process [39,40]. As a result, the intrinsic electrical performance of 2D FETs is significantly degraded.

In this study, the implementation of a buried-gate structure for uniform field induction leads to the improvement of the performance of 2D-MoS_2_ FETs and integrated circuits on a 2-inch scale wafer. By comparing the device performance of local back-gate 2D-MoS_2_ FETs, some parameters showed significant improvements. For instance, the field-effect mobility increased from 0.166 to 1.08 cm^2^/V·s (6.5 times increase), the on/off current ratio increased from 4.90 × 10^5^ to 1.52 × 10^7^ (3.1 × 10^1^ times), and the breakdown voltage increased from 15.73 to 27.48 V, respectively. In addition, it was confirmed that the buried-gate structure showed high uniformity and lower dispersion, while maintaining the on-current state. A high-speed inverter was fabricated using two FETs (a Load FET and a Pull-down FET), and it achieved a DC voltage gain of 17.67 at V_DD_ = 7 V. Moreover, using direct-coupled FET logic technology, more complicated logic circuits (NAND, NOR, AND, OR) composed of three or five FETs were successfully fabricated. In addition, the characteristics of the logic circuits were compared with those of the local back-gate, and the confirmed improvement in performance. Transistors with buried-gate structures will play a key role in the device applications that require improved performance and stability as they are extremely scaled down. Additionally, these preliminary integration efforts demonstrate the promising potential of implementing complex ICs using wafer-scale 2D semiconductors.

## 2. Materials and Methods

### 2.1. Synthesis of MoS_2_ Film on 2-Inch Wafer-Scale SiO_2_ (100 nm)/Si (p++) Substrate

A dry-oxidized SiO_2_ (100 nm)/Si (p++) wafer was cleaned over ultrasonic cleaning for 5 min in acetone, methanol, and isopropyl alcohol (IPA) each. The two-step method was adopted for synthesizing high-quality MoS_2_ film. To synthesize a MoS_2_ film, a layer of MoO_3_ approximately 2.5 nm thick was deposited onto a cleaned 2-inch scale wafer using a Radio Frequency (RF) magnetron sputtering system. MoS_2_ was subsequently synthesized by sulfurizing deposited MoO_3_ using thermal Chemical Vapor Deposition (CVD). During the thermal CVD process, the tube furnace was maintained at a working pressure of 800 Torr. The atmosphere was regulated with a gas mixture of Ar/H_2_S (0.1%) flowing at a rate of 200 sccm. The sulfurization process was initiated by gradually raising the temperature from room temperature to 900 °C over a ramping period. The temperature was maintained at 900 °C for an hour to promote the growth and crystallization of the MoS_2_ film. The synthesis process for the MoS_2_ film is illustrated in Appendix A. Finally, the temperature inside the tube was allowed to naturally cool down to room temperature (R.T.), thus ensuring the preservation of the structural integrity of the MoS_2_ film.

### 2.2. Characterization of MoS_2_ Film

The confirmation of the layered MoS_2_ film structure on the SiO_2_ (100 nm)/Si substrate was conducted using Focused Ion-Beam Transmission Electron Microscope (FIB-TEM). Additionally, the elemental compositions (Mo, S, Si, and O) were determined using Energy-Dispersive X-ray Spectroscopy (EDS) mapping analysis. The Raman spectroscopy of the synthesized MoS_2_ films was measured using a NANOBASE XperRAM-CS Raman spectrometer with 532 nm laser excitation. The components and chemical composition of the MoS_2_ film were examined thoroughly using X-ray photoelectron spectroscopy (XPS) with NEXSA equipment. Atomic Force Microscope (AFM) was utilized to investigate the surface roughness of the MoS_2_ film.

### 2.3. Device Fabrication

A photoresist (PR, AZ 5214) was coated using a spin coater on a SiO_2_ (100 nm)/Si substrate. The coated substrate was soft baked for 50 s at 110 °C using a hot plate, followed by UV exposure (MDS-400 s) for gate patterning. Subsequently, the PR was stripped using a developer (AZ 300 MIF) for approximately 35~50 s and then hard baked for 5 min at 90 °C on a hot plate. The SiO_2_ layer of the gate patterned area was etched using Reactive Ion Etching (RIE) with CF_4_ gas (30 sccm, 100 W, and 2 min). The depth of the etched SiO_2_ layer has been confirmed to be approximately 60 nm. Subsequently, a Ti/Au (5/50 nm) layer was deposited using an E-beam evaporator. To achieve stable device operation, it was necessary to remove the PR residues that remained along the sidewalls of the patterned PR, as well as the spiky protruding residue resulting from the gate-metal process. Then, a 30 nm Al_2_O_3_ layer was deposited as a gate dielectric layer using Plasma-Enhanced Atomic Layer Deposition (PEALD). In order to transfer the MoS_2_ films on to the target substrate, synthesized MoS_2_ film was coated with PMMA (KAYAKU A5). The coated MoS_2_ film was etched in a 5% HF solution for about 30 s, and the separated PMMA/MoS_2_ film was carefully rinsed three times with deionized (DI) water for five minutes each. Subsequently, the PMMA/MoS_2_ film was transferred onto the gate dielectric layer, after placing the wet transferred MoS_2_ film baking at 110 °C for half a day using the hot plate. The wet-transfer process of the MoS_2_ film is shown in Appendix A. In order to remove PMMA, the substrate that was transferred with a MoS_2_ film was removed using acetone and isopropyl alcohol (IPA). Then, an annealing process was performed at 200 °C in a high vacuum (under 10^−6^ Torr) for an hour.

To minimize the PR residue effect on MoS_2_ film, Au masking fabrication was performed for channel patterning [41,42]. After deposited 20 nm of Au, a channel was formed through the photolithography process. The region excluding the channel was selectively etched using a gold etchant. The active channels of the MoS_2_ film were formed using Reactive Ion Etching (RIE), and the residual PR was stripped using acetone. This was followed by annealing under the same conditions as described previously. A 50 nm thick layer of Au was deposited on the source/drain using an E-beam evaporator. The pattern-forming process is the same afterwards. Finally, a passivation layer of Al_2_O_3_, with a thickness of 30 nm, was deposited using Atomic Layer Deposition (ALD). Figure 1a shows an Optical Microscope (OM) image of the integrated MoS_2_ logic circuit on a 2-inch wafer, illustrating its components including inverters, NAND, AND, NOR, and OR gates. The fabrication flow scheme for the MoS_2_ buried-gate inverter is shown in Figure 1b and Appendix A.

### 2.4. Electrical Measurements

The electrical characteristics of the MoS_2_ transistor and various logic devices were comprehensively evaluated using a semiconductor parameter analyzer (Keithley-4200 SCS, Keithley, Cleveland, OH, USA) at room temperature (R.T.). A DC voltage bias was applied to V_DD_ using a Source meter (Keithley-2450). Source Measure Units (SMUs) were connected to V_IN_ and V_OUT_, while the source electrode was connected to GND. The image of the electrical measurement of a logic device using a semiconductor parameter analyzer system is shown in Appendix A.

## 3. Results and Discussion

### 3.1. Characterization of MoS_2_ on the 2-Inch SiO_2_ (100 nm)/Si Wafer

A two-step method was used to achieve the synthesis of high-quality MoS_2_ films. Figure 2a shows the layered structure of the MoS_2_ film for ultrathin-cross sections of materials and their subsequent imaging at nanoscale resolutions on the SiO_2_ (100 nm)/Si substrate, which was confirmed by the focused ion-beam transmission electron microscope (FIB-TEM) image. It provides insights into the internal structure of materials, such as layered structures and interfaces. And, the FIB-TEM tool enables the observation of crystal structures, defects, and nanostructures with atomic-scale resolution [43]. Elemental constituents were identified using Energy-Dispersive X-ray Spectroscopy (EDS) mapping analysis. This method is used in conjunction with electron microscopes such as TEM or SEM to analyze the elemental composition of materials. This is accomplished by detecting the characteristic X-rays emitted as electrons interact with the sample [44]. The synthesized MoS_2_ film demonstrated a three-layer structure characterized by an atomically smooth and flat morphology. The elements Mo, S, Si, and O were confirmed through EDS elemental mapping. The crystal structure of MoS_2_ was attributed to the constituents of Mo and S. Figure 2b shows an image of a uniformly synthesized MoS_2_ film on a 2-inch scale wafer. Figure 2c shows the Raman spectrum of the MoS_2_ film, which was measured at nine distinct points across the 2-inch wafer-scale. Raman spectroscopy is a non-destructive technique that utilizes laser light scattering to analyze the vibrational and rotational modes of molecules in a sample. The identification of molecular species and chemical bonds allows for the identification of materials [45,46,47,48]. Notably, uniform peaks were consistently observed at these locations. The two pronounced peaks at 382.6 cm^−1^ and 403.7 cm^−1^ corresponded to the in-plane vibration (E^1^_2g_) of Mo and S atoms, and the out-of-plane vibration (A_1g_) of S atoms. The difference between these peaks, Δk, indicates the number of layers of the MoS_2_ film. The values exhibited by monolayer and bulk MoS_2_ Δk are 18.9 cm^−1^ and 25.2 cm^−1^, respectively [19,48]. The Δk value of our MoS_2_ film, measured at 21.1 cm^−1^, indicates approximately three layers. The Appendix A shows the actual Raman mapping image of a 2D-MoS_2_ film synthesized on a 2-inch scale wafer. The mapping divides an area of 50 × 50 μm^2^ into sections of 36 points, each with a step size of 10 μm. And, we calculated each of these values ∆k and mapped them to Figure 2d, revealing the values ranging from 20.9 to 21.5 cm^−1^. The components of the MoS_2_ film were analyzed using X-ray Photoelectron Spectroscopy (XPS). This measurement tool uses X-ray irradiation to analyze the surface composition and chemical state of materials by measuring the kinetic energy of emitted electrons. XPS analysis can determine the surface concentration of elements [49]. In Figure 2e, the XPS spectra show the peaks at 232.9 eV and 229.7 eV, corresponding to Mo 3d^3/2^ and Mo 3d^5/2^, respectively. The peak at 226.8 eV represents S 2s. The XPS spectrum corresponding to S 2p, presented in Figure 2f, displayed prominent peaks at 163.6 eV and 162.5 eV, signifying S 2p^3/2^ and S 2p^1/2^, respectively. Figure 2f shows distinct peaks at 163.6 eV and 162.5 eV corresponding to S 2p^3/2^ and S 2p^1/2^, respectively [50,51,52]. Furthermore, the determined composition ratio of Mo and S in MoS_2_ was approximately ~1:2.11. Figure 2g shows an Atomic Force Microscope (AFM) image of the MoS_2_ film. This provides sub-nanometer resolution of 3D topographic information. It quantifies surface roughness and can detect nanoscale features [53]. The analysis was conducted within a chosen area of 10 μm × 10 μm. The Root Mean Square (RMS) roughness, which serves as a measure of surface uniformity, was measured at 0.510 nm. This confirms that our process enables uniformity on a wafer-scale [54].

### 3.2. Comparative Analysis of Electrical Characteristics between MoS_2_-Based Transistors with Buried-Gate and Local Back-Gate

Figure 3a–d show the Atomic Force Microscope (AFM) image and the cross-sectional scanning electron microscope (SEM) image for the buried and local back-gate structure, respectively. The edge of the buried-gate configuration is almost flat. However, the images show that the presence of photoresist (PR) causes the formation of edges in the corner regions, resulting in the development of metallic sidewalls. Appendix A shows the height profiles along with the AFM image. Appendix A shows SEM images of tilted gate surfaces for the each gate structure. The buried-gate is embedded stably in the etched SiO_2_ layer and displays a clean edge shape. In contrast, the local back-gate surface features a sidewall at the edge, which leads to the metal being deposited and rolling out, ultimately resulting in a rough surface. When a voltage is applied to the gate, it causes the voltage distribution to become non-uniform and concentrated along the sidewall, resulting in a faster breakdown voltage. This non-uniformity in voltage distribution leads to inconsistent electrical properties and ultimately has a negative impact on device performance when compared to buried-gate structures. The transfer characteristics and output curves in device performance are the fundamental electrical properties for understanding the working principle of MOSFETs. Also, these device characteristics allow for qualitative and quantitative understanding of intrinsic transistor properties such as mobility and carrier density, interface states and contact resistance. Figure 4a shows the transfer characteristic curves (I_D_ − V_G_) of MoS_2_ field-effect transistors (FETs) with the buried-gate (red line) and local back-gate (blue line) configurations. Notably, the two types of MoS_2_ FET devices share identical structures, including channel dimensions (W/L = 50/10 μm), respectively. In terms of representative data, the buried-gate MoS_2_ FET demonstrates enhanced gate voltage stability and improvement of on-current. The statistical data of the transfer curve of the FETs, according to gate structure, are shown in Appendix A. Figure 4b shows the output characteristics (I_D_ − V_D_) of the MoS_2_ FET. The gate voltage (V_G_) is varied from 2 to 8 V in steps of 2 V, within V_D_ = 3 V. Notably, the buried-gate FET (red line) shows a higher drain current in comparison to the local back-gate FET (blue line). The gate leakage current and breakdown voltage of the gate dielectric are important factors for determining the stability of the device and the electric field distribution of gate dielectric for MoS_2_ FET. Figure 4c shows the gate leakage current (I_G_ − V_G_) behavior of buried-gate and local back-gate FETs. Interestingly, the leakage current over the gate voltage shows the two-step-like behavior indicative of gate dielectric breakdown. For the local back-gate device (blue line), the level of gate leakage current is low, at around 10^−11^ A within a gate voltage of 15 V. Additionally, the gate leakage current increases up to 10^−7^ A from 15 to 20 V of gate voltage, indicating a gradual generation of soft breakdown of the Al_2_O_3_ gate dielectric due to the enhanced gate voltage [55,56,57]. Finally, the gate leakage current rapidly increases to over 10^−3^ A at a gate voltage of 20 V, resulting in the occurrence of hard breakdown of the Al_2_O_3_ gate dielectric. In this stage, the function of the gate dielectric cannot be further accomplished due to the formation of permanent leakage path. On the other hand, for the buried-gate device (red line), the gate voltage ranges for soft and hard breakdown voltage are drastically enhanced up to a gate voltage of 30 V. By utilizing the optimized buried-gate structure of MoS_2_ logic, FETs can achieve enhanced drive on-current and prolonged gate voltage stability simultaneously through uniform gate controllability.

The statistical analysis for various electrical parameters, including the on-off ratio, field-effect mobility, and breakdown voltage, is depicted in Figure 4d–f. Figure 4d,e illustrate the statistical distributions in the on/off ratio and field-effect mobility for buried-gate and local back-gate MoS_2_ FETs. The estimated average values of on/off ratios were found to be 1.52 × 10^7^ (buried-gate) and 4.90 × 10^5^ (local back-gate). The on/off ratio is derived from the comparison of the maximum on-state current with the off-state current to reflect the device’s switching capabilities and suitability for digital circuit applications [58]. Furthermore, the value of field-effect mobility can be deduced from the linear segment of the transfer curve using Equation (1), as below,
(1)$$\mu =\frac{{∆I}_{DS}}{{∆V}_{BG}}\times \frac{L}{WC_{ox}V_{DS}}$$
where μ is the field-effect mobility, I_DS_ is the drain current, V_BG_ is the gate voltage, L is the channel length of 10 μm, W is the channel width of 50 μm, and C_OX_ is the oxide capacitance per unit area between the channel and gate dielectric layer (Al_2_O_3_) [59,60]. The average mobility was calculated to be 1.08 cm^2^/V·s (buried-gate) and 0.166 cm^2^/V·s (local back-gate). The improvement in field-effect mobility for the buried-gate structure is approximately 6.5 times higher compared to the local back-gate structure. Figure 4f presents the statistical variation of breakdown voltage. The breakdown field of the gate dielectric in FETs is a critical design consideration because exceeding this breakdown field can lead to irreversible damage to the device and a breakdown in the insulating properties of the oxide layer. The breakdown voltage of the Al_2_O_3_ gate dielectric layer in FETs can depend on factors such as the oxide thickness, crystal structure, defects, and the fabrication process. Also, Al_2_O_3_ has a high breakdown electric field (5–10 MV/cm) [56,61]. The average values of the breakdown voltage were estimated to be 27.48 V (buried-gate) and 11.75 V (local back-gate), respectively. The breakdown electric field (E_BR_) is defined as V_BR_ divided by T_ox_, where V_BR_ is the breakdown voltage across the insulating layer and T_ox_ is the physical thickness of the insulating layer of Al_2_O_3_ 30 nm. In the breakdown electric field (E_BR_), the buried-gate shows a high value of 9.16 MV/cm, and the local back-gate shows a value of 3.91 MV/cm, which is about 2.3 times lower.

### 3.3. Integrated Logic Circuits Characteristics

After successfully optimizing uniform and high-performance transistors based on buried-gate device configurations, the inverter logic circuit using direct-coupled FET logic was implemented. Direct-coupled FET logic excels in high-speed operation, making it ideal for microprocessors and high-frequency communication systems. It can be optimized for low power consumption, which benefits battery-operated and power-efficient devices. Direct-coupled FET logic is well suited for noisy environments and long-distance communication. The versatility of this device is enhanced by its ability to function with diverse supply voltages across various applications [31,62,63]. In this inverter configuration, the four electrode terminals are fabricated as input gate voltage (V_IN_), output voltage (V_OUT_), drain voltage (V_DD_), and ground (GND). Two neighboring MoS_2_ transistors are concretely formed as load and pull-down transistors. The load transistor acts as a variable resistance, while the pull-down transistor operates as an n-type enhancement-mode FET. The input gate electrode of the pull-down MoS_2_ FET was designated as the input voltage terminal, while the source electrode of the load transistor was connected to the gate electrode via a hole through the Al_2_O_3_ gate dielectric layer, thereby serving as the output terminal of the inverter circuit.

Figure 5b shows the inverter performance of the direct-coupled FET logic configuration by the Voltage-Transfer Characteristics (VTC), where V_DD_ spans from 1 to 7 V with voltage steps of 1 V. The corresponding voltage gain of the inverter was calculated using Equation (2) as below,
(2)$$Gain=-dV_{OUT}/dV_{IN}$$
where d V_OUT_ is the corresponding change in output voltage and d V_IN_ is the corresponding change in input voltage. This equation quantifies how the output voltage changes in response to a change in the input voltage.

The distinct signal inversions are observed with high V_OUT_ in low V_IN_ and vice versa. The corresponding voltage gains (Gain = −d V_OUT_/d V_IN_) are successfully acquired with various V_DD_. The maximum gain was recorded as around 17.67 at V_DD_ = 7 V.

Figure 5c shows the noise margin characteristics at V_DD_ = 5 V. In order to assess the noise margin characteristics of the inverter, we investigated both the low-input signal levels (NM_L_) and high-input signal levels (NM_H_). NM_L_ and NM_H_ were determined by extracting V_OH_, V_OL_, V_IH_, and V_IL_ values at the point where the slope equals −1 on the voltage transfer curve (VTC). It can be quantified using Equation (3) as below [64,65],
(3)$${NM}_{L}=V_{IL}-V_{OL}\;{NM}_{H}=V_{OH}-V_{IH}$$

At V_DD_ = 5 V, the inverter calculated NM_L_ and NM_H_ values as 2.24 V and 0.9 V, respectively. The characteristics of inverters made with a local back-gate are shown in Appendix A.

We further varied the sizes of FETs incorporated into the inverters, observing diverse voltage gains ranging from 1 V to 7 V, and confirmed the shift of the switching threshold voltage. Figure 5d shows the Voltage Transfer Characteristics (VTC) of the inverters, which were evaluated for different ratios of load transistor and pull-down transistor widths, specifically 120/50, 160/50, 200/50, 240/50, and 280/50 μm. All ratios had identical channel lengths of 10 μm. Furthermore, a substantial width ratio between the pull-down and load FETs enhances noise immunity while also shifting the switching threshold voltage to the right. This phenomenon indicates a significant impact on the inverter’s switching behavior, thus illustrating the interconnection between transistor dimensions and their effects on both voltage gain and threshold voltage [66,67].

Finally, based on buried-gate FETs and inverter characteristics, we fabricated a series of complex logic devices composed of three or five MoS_2_-film transistors. Figure 6 shows images of logic circuits using Optical Microscope (OM), consisting of (a) NAND, (b) NOR, (c) AND, (d) OR. The images are displayed on the left, while the corresponding timing diagrams are presented on the right. The channel sizes of the load FET and pull-down FETs are L = 10 μm, W_IN_ = 120 μm, and W_OUT_ = 50 μm, respectively. The gate terminals of two pull-down transistors were connected to the two input terminals, with input voltages of −5 and 10 V corresponding to logical states ‘0’ and ‘1’, respectively. The timing diagrams, which illustrate the buried-gate-based logic circuits, confirm the successful implementation of Boolean expressions [68,69]. They clearly differentiate between a high-output voltage (approximately 3 V) and a low-output voltage (nearly 0 V). In contrast, logic circuits based on the local back-gate were also achieved, but their output state was approximately 2 V. This deviation from the anticipated 3 V is attributed to the negative threshold voltage (V_th_) characteristic of MoS_2_ transistors. The negative V_th_ leads to an inverter output voltage below V_DD_ at 0 V, causing a delayed dynamic switching behavior. Consequently, logic circuits based on buried-gate, when compared to local back-gate, exhibited a more relaxed and faster response speed.

## 4. Conclusions

We demonstrated the advantages of the buried-gate structure through comparative study with the local back-gate structure. The MoS_2_ transistors with the buried-gate structure show enhanced electrical characteristics compared with the local back-gate structure (higher on-state current and on/off ratio at 10^7^, elevated field-effect mobility of 1.08 cm^2^/V·s, and improved breakdown voltage of 27.48 V). In addition, high-quality MoS_2_ film was synthesized using the two-step method. Inverters and logic circuits were successfully fabricated using a buried-gate structure by applying MoS_2_ film to a 2-inch wafer-scale. The inverter demonstrated high voltage gain and noise tolerance, while the logic circuits (NAND, NOR, AND, OR) produced the expected Boolean results. These findings underscore the potential of MoS_2_-based buried-gate devices in advancing logic circuits and integrated circuits.

## Figures and Tables

**Figure 1 nanomaterials-13-02870-f001:**
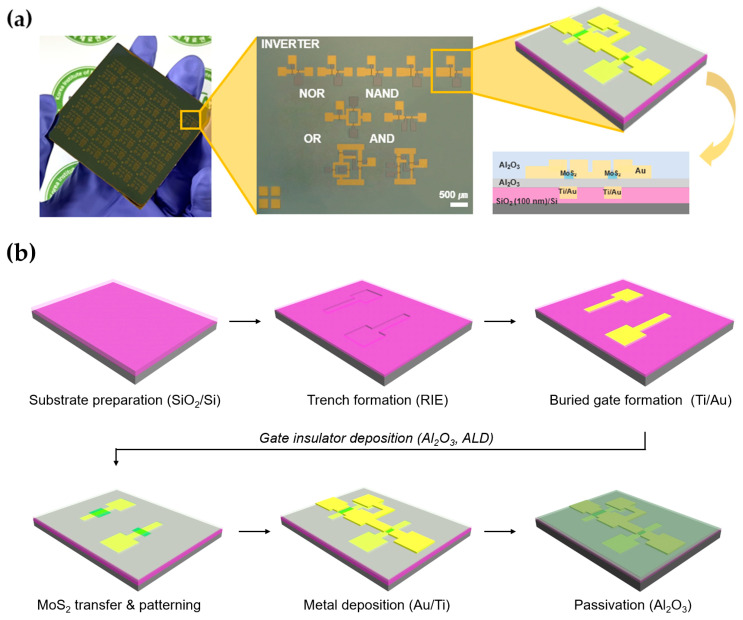
(**a**) Optical Microscope (OM) image and schematic of buried-gate-based logic device on 2-inch wafer-scale SiO_2_ (100 nm)/Si substrate. (**b**) Schematic of main fabrication process of 2D-MoS_2_ inverter using buried-gate structure.

**Figure 2 nanomaterials-13-02870-f002:**
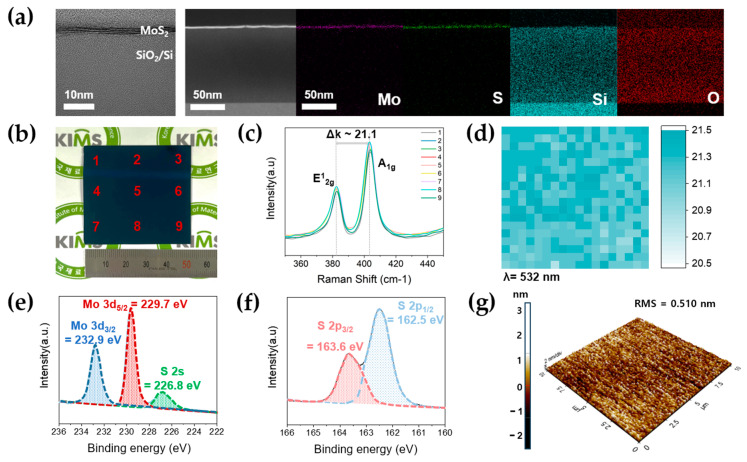
(**a**) Focused Ion-Beam Transmission Electron Microscope (FIB-TEM) image of transferred MoS_2_ film on 2-inch wafer-scale SiO_2_ (100 nm)/Si substrate and mapping of Energy-Dispersive X-ray Spectroscopy (EDS) analysis of MoS_2_ film. (**b**) The image of uniformly synthesized MoS_2_ film on 2-inch scale wafer. A total of 9 Raman mapped locations are indicated. (**c**) Raman spectrum of MoS_2_ film on 2-inch SiO_2_ (100 nm)/Si wafer about 9 points. (**d**) The mapping image of the MoS2 film represents a value that is calculated from Δk of each data point’s individual value. The mapping divides an area of 50 × 50 μm^2^ into sections of 36 points, with each section having a step size of 10 μm. (**e**,**f**) X-ray Photoelectron Spectroscopy (XPS) spectrum of MoS_2_ film. (**g**) The Atomic Force Microscope (AFM) image of MoS_2_ film.

**Figure 3 nanomaterials-13-02870-f003:**
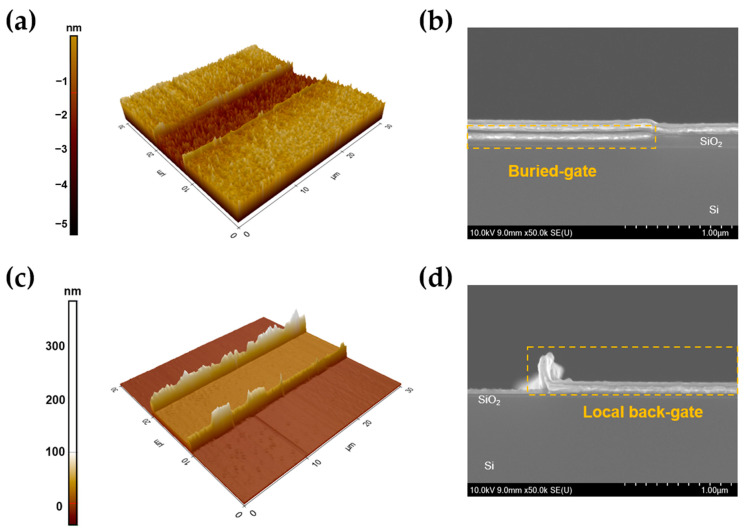
The Atomic Force Microscope (AFM) images of (**a**) buried-gate and (**c**) local back-gate. The cross-sectional scanning electron microscope (SEM) images of (**b**) buried-gate and (**d**) local back-gate.

**Figure 4 nanomaterials-13-02870-f004:**
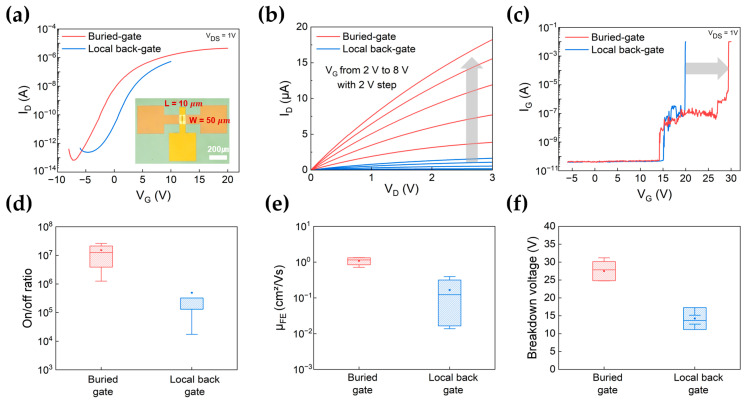
(**a**) Transfer characteristics (I_D_ − V_G_) of MoS_2_-based FET (red line of buried-gate, blue line of local back-gate), the channel length of 10 μm at a drain voltage (V_DS_) of 1 V. (**b**) Output curve (I_D_ − V_D_) of at gate voltage (V_G_) 2 to 8 V (step of 2 V) (red line of buried-gate, blue line of local back-gate). (**c**) Gate leakage curve (I_G_ − V_G_) of MoS_2_ FET (red line of buried-gate, blue line of local back-gate). Statistical histograms of (**d**) on/off current ratio, (**e**) field-effect mobility, and (**f**) breakdown voltage for buried-gate and local back-gate.

**Figure 5 nanomaterials-13-02870-f005:**
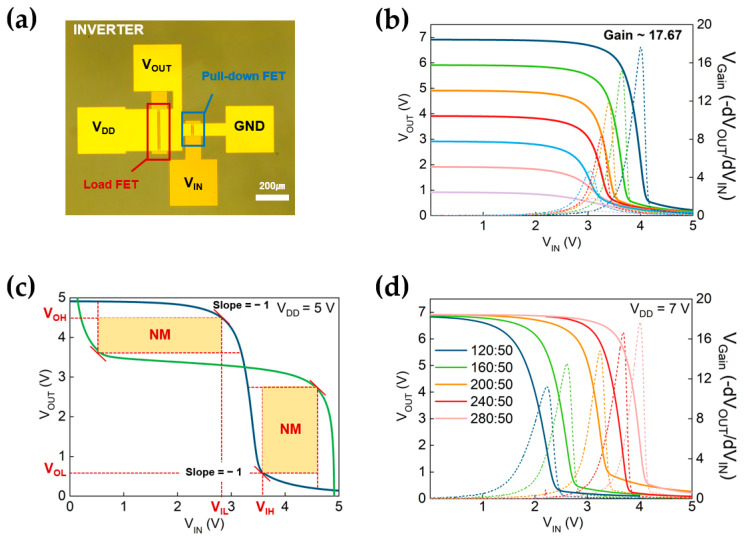
(**a**) An Optical Microscope (OM) image of the inverter based on two n-type MoS2 FETs (Load FET and Pull-down FET). (**b**) The Voltage Transfer Curve (VTC) of the inverter is observed with supply V_DD_ from 1 to 7 V, resulting in the best performance of a voltage gain of 17.67. (**c**) Noise Margin (NM) properties of the inverter at V_DD_ = 5 V, the curve acquired by mirroring the Voltage Transfer Curve (VTC), and the position of slope equals to −1 marked (V_OH_ represents the minimum high-output voltage when the output level is logical “1”; V_OL_ represents the maximum low-output voltage when the output level is logical “0”; V_IL_ represents the maximum low-input voltage, which can be interpreted as logical “0”; V_IH_ represents the minimum high-input voltage, which can be interpreted as logical “1”). (**d**) Voltage Transfer Curves (VTC) at V_DD_ = 7 V of the inverters with different ratios of load transistor and pull-down transistor width of channel (120/50, 160/50, 200/50, 240/50, and 280/50 μm), and identical channel length of 10 μm.

**Figure 6 nanomaterials-13-02870-f006:**
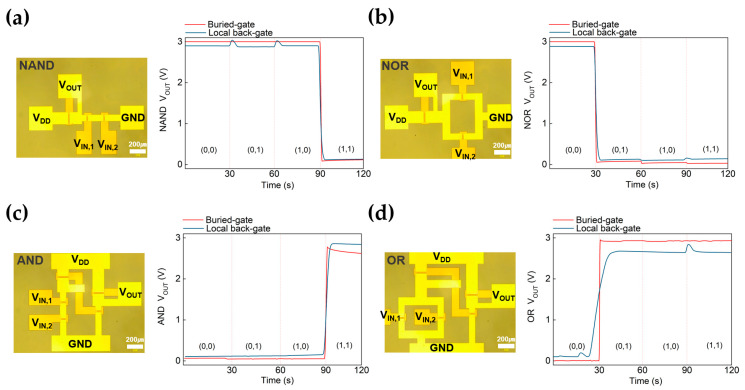
Optical microscope (OM) images (left) and Timing diagrams (right) of (**a**) NAND, (**b**) NOR, (**c**) AND, (**d**) OR gates with comparison between buried-gate (red line) and local back-gate (blue line) at V_DD_ = 3 V. The four typical input states (0, 0), (0, 1), (1, 0), and (1, 1). “0” and “1” of input states represent and −5 and 10 V for input voltage in actual measurement. V_OUT_ results of logic circuits separate the low and high binary output states.

## Data Availability

Not applicable.

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
