# Peer review of "Integrated Logic Circuits Based on Wafer-Scale 2D-MoS2 FETs Using Buried-Gate Structures"

_nanomaterials, 2023, doi:10.3390/nano13212870_

Round 1
Reviewer 1 Report
Comments and Suggestions for Authors
In this paper, the authors presented their buried-gate MoS2 transistor. Compared to the local back-gate structure, the buried-gate transistor showed 6.5 times higher mobility, 31 times higher on/off ratio, and 75% higher breakdown voltage because of less topography.
But unfortunately, all these improvements are mainly because they chose a bad control. From fig3 b, it is very obvious that the local back-gate, their control, has very tall rabbit ears (the metal sidewall) which will degrade the device's performance. (It will be more obvious if they have an SEM cross-section) This is a very common defect due to bad lift-offs and can be easily fixed by using bi-layer lift-offs or etch patterning.
So the authors should first fix the rabbit-ear issue and then use the improved local back gate device as the control. Only by doing so can they tell the actual benefit of the buried gate structure.
Author Response
First of all, we would like to thank you for giving the opportunity as a revised submission “Integrated Logic Circuits Based on Wafer-scale 2D-MoS2 FETs using Buried-Gated Structures”. Once we have had sufficient time to carefully consider all of the reviewer’s concerns, we were able to further improve the quality of the revised manuscript. Thus, we provided and have made major changes as a full point-by-point response according to the reviewer’s suggestions as follows.

Reviewer 2 Report
Comments and Suggestions for Authors
This work used buried-gated structure to improve the performance of MoS2 FETs, which is meaningful.
(1) the gate insulator in the device is Al2O3. Compare with the traditional dielectric material SiO2, Al2O3 will influence the properties of the MoS2 FETs or not?
(2) The paper needs to include the imagination of MoS2 on Al2O3.
Comments on the Quality of English Language
Minor editing of English language required.
Author Response

(The authors gave the same response as above.)

Reviewer 3 Report
Comments and Suggestions for Authors
Overall, the manuscript provides a comprehensive study on the optimization and characterization of MoS2 films. However, enhancing the clarity in some areas and expanding the discussion on the implications of the results would make the paper more impactful.
An abstract provides a concise summary of the research. Ensure the abstract is well-structured and encompasses the main objectives, methods, results, and conclusions.
It would be beneficial to provide some background on the significance and applications of MoS2 films in nanoelectronics to provide context for readers.
It would be helpful to provide the rationale or advantage of using FIB-TEM, EDS, Raman Spectroscopy, XPS, and AFM for the characterizations.
The device fabrication process is detailed but consider adding flow diagrams or a schematic illustration to visualize complex processes better.
The term "fished up" seems colloquial. Consider rephrasing it to a more technical term like "transferred" or "aligned."
While the characterization results are mentioned, a more in-depth discussion on the implications of these results concerning the device's potential performance would enhance the section.
You mentioned the film demonstrated a three-layer structure. It would be beneficial to discuss the implications or advantages of this tri-layer configuration.
Please provide the actual Raman mapping figure or ensure it's included in the final version.
Figures referenced should be scrutinized to ensure they effectively convey the described results. Clear annotations and captions will benefit the reader.
Consider adding electron microscopy images to showcase the PR edges and the mentioned metallic sidewalls.
It might be helpful to provide a brief description of the importance of transfer characteristics and output curves in device performance.
It would be beneficial to clarify the importance or advantages of the 'direct-coupled FET logic' over other configurations, if any.
Consider adding a section or paragraph discussing potential future research directions based on your findings. This can guide other researchers in the field.
Comments on the Quality of English Languageminor
Author Response

(The authors gave the same response as above.)

Reviewer 4 Report
Comments and Suggestions for Authors
The article “Integrated Logic Circuits Based on Wafer-scale 2D-MoS2 FETs 2 using Buried-Gated Structures” is well-written and all the statements are well supported by experimental results. For these reasons, I consider that the article fits in the scope of the journal and can be appropriate for publication. Before that, minor revisions are needed:
1. In the abstract the authors talk about the degradation of the electrical characteristics. Hence, I would like to see if it is possible, experiments that support the electrical stability in the time of the properties of the fabricated FETs.
2. References 1-5 and relatively old. Graphene is a booming field, and the most recent references should be included.
3. In the introduction section, the advantages of this kind of FETs with respect to the most famous OFETs in these days should be included, or a little comparison between their properties.
4. In lines 92-93, 900ºC is a very high temperature for a synthesis. Is the only methodology or why the authors chose this procedure for this synthesis?
After these minor suggestions, I consider that the article can be published in Nanomaterials.
Author Response

(The authors gave the same response as above.)

Reviewer 5 Report
Comments and Suggestions for Authors
Authors have presented the work on the local back and buried gate configuration with the MoS2 channel and their function analysis comparison. Further, authors have demonstrated and integrated buried-gated MoS2 transistor array, integrated logic circuits including inverters, NAND, NOR, AND, and OR gates, with successfully fabrication over a 2-inch wafer scale. The work is within the scope of the MDPI Nanomaterials and the targeted special issue. Reviewer has the following major comments to be addressed.
The introduction is somewhat convoluted and lacks a clear structure.
There are several grammatical issues, such as missing articles (e.g., "the graphene", "the" before "high electron mobility"). and comma splices. Proper punctuation and sentence structure would enhance readability.
Some sentences are lengthy and complex, which can make the text a bit challenging to follow. Consider breaking them into shorter sentences for better readability.
Line 49-54, should be more concise with specific highlight of properties and applications with recent literature such as photodetectors [https://doi.org/10.1016/j.mtnano.2023.100382], optoelectronic memories [https://doi.org/10.1016/j.jpcs.2023.111406], sensors [https://doi.org/10.3390/nano13182502] etc.
Provide the 532 nm laser incident power and spot size used for Raman measurements.
Mention, the reason behind the gate voltage stability and high on current value with buried gate?
What is the thickness of the MoS2 thin film, authors can extract it from the cross-sectional SEM image?
Figure5b,d, right side y-axis should be corrected, with (-dVout/dVin).
What is the level of stability exhibited by the fabricated logic gate device?
Acronyms should be maintained throughout the manuscript.
Author Response

(The authors gave the same response as above.)

Round 2
Reviewer 1 Report
Comments and Suggestions for Authors
The authors have made appropriate adjustments to the manuscript. The manuscript can be accepted in its present form
Reviewer 5 Report
Comments and Suggestions for Authors
Authors haved addressed all the comments. The manuscript can be accepted for publication.